# Experimental Study on the Mechanical Behavior of Dry Corn Stalk Cutting

**DOI:** 10.3390/ma16083039

**Published:** 2023-04-12

**Authors:** Dominik Wilczyński, Krzysztof Wałęsa, Krzysztof Talaśka, Dominik Wojtkowiak

**Affiliations:** Faculty of Mechanical Engineering, Institute of Machine Design, Poznan University of Technology, Piotrowo Str. 3, 60-965 Poznań, Poland; krzysztof.walesa@put.poznan.pl (K.W.); krzysztof.talaska@put.poznan.pl (K.T.); dominik.wojtkowiak@put.poznan.pl (D.W.)

**Keywords:** cutting parameters, biomass, optimization

## Abstract

This work presents an experimental study of cutting corn stalks for thermal energy generation. The study was carried out for the values of blade angle in the range of *α* = 30–80°, distance between the blade and the counter-blade *g* = 0.1, 0.2, 0.3 mm and the velocity of the blade *V* = 1, 4, 8 mm/s. The measured results were used to determine shear stresses and cutting energy. The ANOVA variance analysis tool was used to determine the interactions between the initial process variables and the responses. Furthermore, the blade load-state analysis was carried out, together with determining the knife blade strength characteristic, based on the determination criteria for the strength of the cutting tool. Therefore, the force ratio *F_cc_*/*T_x_* was determined as the determinant of strength, and its variance characteristic in the function of the blade angle, *α*, was used in the performed optimization. The optimization criteria entailed the determination of such values of the blade angle, *α*, for which the cutting force value, *F_cc_*, and the coefficient of knife blade strength approached the minimum value. Hence, the optimized value of the blade angle, *α*, within the range 40–60° was determined, depending on the assumed weight parameters for the above-mentioned criteria.

## 1. Introduction

Biomass constitutes the most important renewable source of bound carbon, and may be reprocessed into liquid fuels as well as solid and gas fuels together with other chemical products, apart from serving as a source of heat and electrical energy [1].

The term biomass is applied to all organic materials originating from plants which can be converted into energy [2]. Biomass can be burned directly in power plants, converting waste to energy, without additional chemical processing, in order to generate steam and electrical energy [3]. In the process of converting biomass into energy through combustion, compacted or bulk (loose) biomass is typically used. In both cases, it may be necessary to reduce the dimensions of the biomass particles through cutting or shredding. Therefore, this process, or more accurately, its energy consumption, should be accounted for in the overall energy balance calculations for generating energy from biofuels. It should be considered as one of the initial stages affecting the overall efficiency and economic feasibility of the process of generating compacted biofuel from waste biomass. The reduction in particle size facilitates the transmission of heat and mass in the course of the compaction process, directly improving its efficiency. This further results in lower energy consumption of the process, leading to improved economic feasibility [4].

The parameters of the cut or fragmented biomass directly affect the efficiency of the compaction process, as well as the mechanical and energy characteristics of the resulting briquette or pellet. This phenomenon has been accounted for in numerous studies [5,6,7]. The purpose is to achieve the highest possible energy concentration together with the highest possible resistance to mechanical damage.

In order to obtain the desired characteristics of biofuel obtained in the compaction process utilizing the piston or worm screw [8,9,10,11,12,13], it is necessary to modify the physical and chemical characteristics of loose biomass, in order to improve its capability to undergo the process of agglomeration. The primary concern is to select the correct particle size. Numerous researchers have studied the cutting and fragmentation processes of biomass, more exactly the effect of different process parameters such as blade geometry, the kinematic system of the cutting device, the moisture content of the cut biomass, the degree of its deformation, etc.

Thus, a comparative study by Zastempowski and Bochat focused on the performance of the classic and new design of cutting drums. The study entailed cutting rye straw into sections of a defined length. The classic drum construction was cylindrical, positioned across the cut material. In contrast, the new design was based on a double truncated cone. This design facilitates the performance of two slanted cuts in two directions. The authors performed the study with four variants of the construction of the cutting drum, the cutting angles being, respectively, *α* = 0°, 15°, 30° and 45°. The values used to evaluate the process were output and energy consumption per unit, as well as cutting resistance per unit. The results indicate that the new drum design allowed for improvement in the process output of 25%, a reduction in the energy consumption per unit by up to 34%. The cutting resistance per unit was reduced to 8% [14].

A study by Van-Dam Vu et al. examined the cutting process of corn stalk, which led to determining the optimized parameters for both the cutting force and power, considering the blade velocity of 4.40 m/s, a blade-angle value of 30° and a rake angle of 50°. The determined cutting-force value for the above parameters was equal to 251.7 N, and the power was equal to 26.2 W [15].

Abdellatif Barakat et al. performed a study of the fragmentation process of wheat straw, rice straw, corn stalks and blooms, as well as sorgo and silvergrass, examining the influence of moisture content and chemical composition of the material on the process. In the process of fragmentation of wheat straw from an initial straw length of 300–600 mm to below 0.25 mm, a reduction in process energy consumption was observed after reducing the initial moisture content in the straw from 7% to 1% [16].

A study by Venkata S.P. Bitra et al. focused on the fragmentation of wheat straw, switchgrass and corn stalks. They examined different sieve dimensions of the knife mill, rotational speed and feed speed of the material intended for fragmentation. The total specific energy for the fragmentation of switchgrass, wheat and corn stover increased with the rotational speed of the knife mill. This energy decreased together with the increase in the screen size for each examined biomaterial. Similarly, its value decreased together with the increase in biomass feeding speed. For the knife mill sieve size of 25.4 mm and the optimal speed of rotation equal to 250 revolutions/min, the optimal material feeding speeds were, respectively 7.6, 5.8 and 4.5 kg/min for switchgrass, wheat straw and corn stover, and the respective specific energy values were equal to 7.57, 10.53 and 8.87 kWh/Mg [17].

Mohsen Azadbakht et al. examined the energy consumption of the cutting process of canola stems performed at different heights of the stem. The cut stems were characterized by different levels of moisture content. The testing was repeated 15 times for each level of moisture content and height. Results indicate that the effect of the height of the cut and moisture content on cutting energy is significant, but the correlation between these parameters is not statistically significant. The highest value of energy used, 1.1 kJ, was observed when cutting stems with a moisture content of 25.5%, and with the cut performed at a height equal to 10 cm. The lowest value of used energy was equal to 0.76 kJ for the stem moisture content of 11.6% and the cut performed at a height equal to 30 cm. The linear velocity of the blade during the cutting process was equal to 2.64 m/s [18].

Certain researchers were interested in modelling the cutting process. A work by Zastempowski and Bochat presents a new mathematical model representing the cutting process of plant material utilizing a scissor-finger cutting system. The model accounts for all the stages of the shearing process: the deformation along the stem cross-section and the separation of the stem into two sections. Furthermore, the model accounts for the variable stem rigidity along its length. The authors employed the described model in a series of simulations. The results demonstrate a high degree of conformity with the experimental results. The model may be employed at the design stage of the cutting device. Mechanisms of this type are used in combine harvesters, field straw cutters and mowing machines. The results of the study constitute a basis for introducing alternative construction solutions to replace existing ones that are characterized by high energy consumption [19].

Gao et al. examined the cutting process of the Caragana korshinskii plant stems, seeking to identify the relations between the diameter of the stem, the blade angle, rake angle, blade velocity, distance between the blade and the counter-blade (in the case of supported cutting), the height of the cut and the moisture content in the stem. The study considered two variants of the cutting process: employing a single blade, without supporting the stem (unsupported cutting) and cutting employing the blade and the counter blade supporting the blade (supported cutting). The variance analysis of the obtained results (ANOVA) leads to the conclusion that for unsupported cutting, the best combination of parameters includes a blade velocity of 3.315 m/s, a blade angle of 20°, and a rake angle of 20°, for which the cutting force value is equal to F = 95.690 N. For supported cutting, the parameter values are respectively: 3.36 m/s, 20°, 20°, and the distance between the blade and counter-blade is equal to 1.38 mm, for which the cutting force value is F = 53.082 N [20].

A further study by Gao et al. focused on the supported cutting process of the Caragana korshinskii plant; the single-parameter test demonstrates an increase in the cutting force as well as the specific energy of the cutting process test, together with the increase in the stem diameter, and the values decreased as the moisture content in the cut material decreased. The increase in the blade-angle value in the range of 20–35° caused an increase in the cutting force value and the specific energy of the cutting process showed a downward trend. For the blade rake angle in the range of 0–20°, the force and cutting-energy values decreased. The multi-parameter Box–Behnken test allowed for the optimization of the model of force and specific energy of the cutting process. This allowed for the determination of the optimal parameter values for this process, that is: knife velocity 0.5 m/s, blade angle 25°, rake angle 20° and a distance between the blade and the counter-blade of 1.4 mm. For these parameters the cutting-force value was F = 644.38 N [21].

Van-Dam Vu et al. proposed their own design of a device for cutting corn stalks. It was employed in the testing of the cutting process of corn stalks with a moisture content equal to 81%, and the purpose thereof was to determine the cutting-force value and the average power consumption. Based on a multicriteria optimization of the obtained results, it was ascertained that the correct selection of process parameters may cause a reduction in the cutting force and energy consumption by a factor of two, three, and up to four. The energy consumption in the course of the cutting process may be significant, even with the low value of the cutting force. This occurs when the knife velocity is high. It is therefore desirable to minimize the value of the cutting force, together with the energy consumption [15].

A study of the cutting process of cabbage seedlings employing a counter-blade (supported cutting) was carried out by Wang et al. [22]. The obtained results indicate that with the increase in the value of the blade rake angle, the cutting stresses and specific energy would initially decrease, and afterwards their values would increase. The increase in the blade angle was conductive to the decrease in the cutting stress and specific energy. The increase in the distance between the blade and the counter-blade caused an increase in the value of the previously mentioned parameters. The ANOVA variance analysis (response surface) indicates that the significant factor influencing the maximum cutting stress was the blade angle, together with the distance between the blade and the counter-blade.

Zhang et al. [23] examined the process of cutting rice stems with blade-angle values of 0°, 30°, 45° and 60°. It was determined that the correct selection of the blade sharpening angle and the height of the cutting point has a significant benefit on energy saving. The average energy required to separate the stem increases as the point of the cut is removed from the ground level. Peak cutting force per unit of measurement of the stem cross-section decreases together with the increase in the blade sharpening angle. The determined optimal blade-angle value is 45° for cutting without a counter-blade (unsupported cutting) and 30° for cutting with a counter-blade (supported cutting).

Beneficial findings from the standpoint of practical application in the design and development of robotic shears for cutting citrus saplings were achieved by Xie et al. [24]. Based on the variance analysis of the process parameters and the obtained response, they developed optimized working parameters for the process, i.e., blade angle of 27°, blade rake angle of 50°, blade velocity equal to 285 mm/min and blade thickness of 2.83 mm. The above parameter values result in an average cutting force equal to 747.98 N and a cutting energy of 55.48 mJ/mm^2^. The authors confirm that the optimization of the geometric parameters of the blade result in an increased cutting quality and reduced energy consumption.

Wilczyński et al. [25] determined the lowest cutting force for triticale straw in a drum-knife with a helical blade geometry, for a blade angle of 45° and a rake angle of 15°, with a lower value of the stem moisture content equal to 16.94%.

During an examination carried out on an MTS testing machine which involved cutting a single stalk of the triticale straw, the lowest recorded cutting strength was for a blade angle of 45° and a rake angle of 30°, at a material moisture content of 10.5% (Wilczyński et al.) [26].

Wałęsa et al. [27] recorded the lowest value for the cutting force for the distance between the blade and the counter-blade equal to 0.1 mm and blade velocity of 8 mm/s during an examination of the cutting process of triticale straw with a moisture content of 14.8%, using a knife with a blade angle of 90° and a raking angle of 0°.

The analysis of the current state of scientific knowledge and ongoing research indicates a high degree of significance of the studies on the influence of the cutting (fragmentation) process parameters on energy consumption. The cut biomass at the specified particle size can be, e.g., burnt or subjected to agglomeration in order to produce biofuel. Hence, the energy consumption of the cutting process is a constituent of the overall energy consumption for the process of manufacturing biofuel or generating energy, e.g., in the process of biomass combustion to generate thermal energy; hence, it is a very important factor in the final economic feasibility of the energy generation process from the standpoint of its final balance, ignoring the ecological factors, which are the most important consideration.

Corn stalks constitute a very good source of fuel for burning, and are characterized by double the calorific value, with a reduced moisture content [28,29].

The works cited above examine the influence of the cutting process parameters on its energy consumption, the power requirement to carry out the process and the shear stresses, as well as the quality of the product. The main parameters whose influence has been studied were the knife blade angle, the rake angle, the distance between the blade and the counter-blade, knife thickness, knife linear velocity and the moisture content of the cut material, as well as the geometric parameters of the cross-section at the cutting point. The studies employed different types of biomaterials, which are typically available in a given region where the authors’ research unit is located. Similarly, the authors of the present manuscript also employed locally sourced material in the examination of the cutting process, i.e., the corn stover. The common characteristic of the studies is primarily the evaluation of the effect of the geometric parameters of the knife, i.e., the blade angle, the rake angle and the distance between the blade and the counter-blade. Therefore, the influence of these parameters, apart from the rake angle, is the focus of the authors of the presented work. Hence, the presented study considered an expanded range of blade angles to include the values *α* = 30–80° in increments of 10°. The authors’ goal was to accurately determine the influence of the blade angle in a broader range of values, which may provide a more comprehensive characteristic of variance for, e.g., the energy consumption of the cutting process. An interesting and novel approach by the authors of the present work is to employ a correction of the negative effect of the lack of uniformity of the cross-sectional area of the cut material on the dispersion of the cutting force Fc values recorded in the course of the experiment, as well as on the other determined parameter values such as the ultimate shear stress, *τ*, total cutting energy, *E_tce_*, and specific cutting energy, *E_sce_*. This result was achieved through measuring the cross section of the stem (see Section 3, Figure 6). The dispersion of the measured cross-section of the stems was the basis for correcting the abovementioned parameters and consequently reducing the negative effect of the stem cross-sectional dispersion on the determined dependencies between the input parameters of the cutting process (setpoints) and the responses. In the authors’ opinion, this constitutes a reliable tool for correcting the results obtained in the study, allowing the mitigation of the error introduced by the lack of uniformity of material used in this type of study. The function employed for the correction is provided in Section 3, Equation (1).

The results of the study were subject to ANOVA variance analysis. The resulting conclusion is that the key process variable is the knife blade angle, *α*, (see Section 4, Table 7). The influence of the distance between the blade and the counter-blade, *g*, as well as the knife velocity, *V*, was significantly lower. A component of potential interest in the present study is the analysis of the distribution of forces on the blade (Section 4.1, Figure 11). This was used to determine the value of the ratio *F_cc_*/*T_x_*, used as a criterion for the blade life. A characteristic curve to describe the variance of this parameter as a function of the blade angle *α* was developed. It was employed as a basis for optimization (see Section 4.2). An interesting issue of the optimization presented in Section 4.2 is the analysis of variance of the optimum values of the blade angle, *α*, together with the distance, *g*, and velocity, *V*, depending on the assumed weights for the individual optimization criteria (see Table 9).

## 2. Materials and Methods

### 2.1. Material Preparation

The material used in the study was corn harvested from the fields located in the Wielkopolska Province (west-central Poland) at the following geographical coordinates: 52°00′08.0″ N 17°46′46.9″ E. The corn stalks underwent seasoning for a period of 12 months after being harvested, by placing them indoors at room temperature. The moisture content of the stalks was determined by using a Mettler Toledo scale-dryer. The average moisture content from ten measurements was 9.01%, see Figure 1.

### 2.2. Tests

The study of the cutting process was carried out using the testing station shown in Figure 2. It consists of an upper plate, 3, directly affixed to the jaws of the testing machine, 6, moving along the vertical roller guides, 5. The use of roller guides, 5, facilitates precise control over the upper plate, 3, and consequently of the knife, 1, with a view of maintaining a constant distance between the knife blade, 1, and the counter-blade, 2 (see Figure 3). In the course of the study, the upper plate, 3, performs a reciprocating motion forced by the linear motion of the jaws of the testing machine, 6. The cutting knife, 1, is affixed directly to the upper plate, 3. The counter-blade, 2, is affixed to the bottom plate, 4. The corn stalk, 7, is placed between the blade, 1, and the counter-blade, 2, and the test begins, entailing the linear downward motion of the jaws of the testing machine, 6, causing an equal motion of the upper plate, 3, affixed to the jaws and the knife blade, 1, affixed to the plate. This causes the knife blade, 1, to cut through the corn stalk, 7. In the course of the study, the value of the cutting force, *F_c_*, on the corn stem, 7, was recorded, together with the displacement indicated by the sensors of the MTS testing machine. This measurement was carried out for the varied blade angle, *α*, the distance between the blade, 1, and the counter-blade, 2, referred to elsewhere in the present work as *g* (Figure 3), as well as the linear velocity of the blade, 1, denoted by *V*. The successively used blade-angle values *α* were *α* = 30°, 40°, 50°, 60°, 70° and 80°. The values of the distance between the blade, 1, and the counter-blade, 2, were equal to *g* = 0.1, 0.2, and 0.3 mm, and the values of the knife blade linear velocity were *V* = 1 mm/s, 4 mm/s and 8 mm/s. For each value of the blade angle *α*, the effect of each value of the distance between the blade, 1, and the counter-blade, 1, *g* was examined, together with each velocity value, *V*. For each set of initial parameters of the cutting process, *α*, *g* and *V*, the test was repeated ten times. In this fashion, 540 test cuts of the corn stalks were carried out.

The general view of the spacing between the blade, 1, and the counter-blade, 2, of the testing station, as well as the knives employed in the testing, is provided in Figure 3.

## 3. Results

The cutting process can be divided into three stages (see Figure 4 and Figure 5). Figure 4 presents an example graph line for the recorded force value *F_c_* in the course of the study of the corn-stalk cutting process. The first stage entails compression of the corn stalk by the blade. It shows a mild and proportional increase in force, along with the linear displacement of the knife. In the second stage, a sudden decrease in force is observed, by approx. a dozen Newtons (see Figure 4, marked with an ellipsis), followed by a sudden increase in the *F_c_* value, achieving the maximum with a small displacement of the knife as it penetrates the previously compressed material of the corn stalk. The third and final stage is characterized by a sudden decrease in the force value, *F_c_*, caused by the final separation of the corn stalk material (see Figure 4 and Figure 5).

The results obtained in the course of the experiment were subjected to an ANOVA variance analysis for variable input parameters (*α*—blade angle, *g*—distance between the blade and the counter-blade, *V*—linear velocity of the blade) and the responses determined on the basis of the recorded values, i.e., the cutting force, *F_c_*. The ANOVA variance analysis enabled us to seek the interdependences between the input parameters mentioned above (the input variables of the experiment) and the responses achieved in this experiment. This allowed us to obtain the functional dependencies of the variances in the experimental responses and their effect on the cutting process variables. The determined value of the corn stalk cross sectional area *A* prior to the study of the cutting process was used to correct the value of the measured cutting force, *F_c_*. This is caused by the variance of the cross-sectional area of the corn stalks, and the introduced correction allows for the decrease in the negative influence of this variance on the measured cutting force value, *F_c_,* as well as on the values of all other process parameters determined post-correction, i.e., the following responses: ultimate shear stress, *τ*, total cutting energy, *E_tce_*, specific cutting energy, *E_sce_*. Table 1 presents the example values of the corn stalk cross-sectional area (an elliptical shape) which were determined for the blade angle *α* = 30°, and the distances *g* = 0.1, 0.2, and 0.3 mm, as well as the blade velocities *V* = 1 mm/s, 4 mm/s and 8 mm/s.

Each cross-sectional area was determined using the AutoCad software. First, a photograph was taken for a given cross-section after the test, next to the scale of a slide caliper to obtain a size reference. The photograph was then loaded onto the AutoCad software (see Figure 6) and resized to actual dimensions, using the size reference. On the photograph, the outer edge of the corn-stalk cross section was highlighted, using the spline line function (see Figure 6). Afterwards, the measure area function was used to determine the cross-sectional area of the stalk.

The above table clearly demonstrates that the cross-sectional area of the corn stalk varies, and therefore it is recommended to introduce a correction to the value of the cutting force, as per the following Formula (1):(1)$$\begin{array}{c} F_{cc}=\left\{\left[\left|\left(\frac{A_{1}+A_{2}+\ldots +A_{n}}{n}\right)-A_{n}\right|÷A_{n}\right]\cdot F_{cn}\right\}+F_{cn};\;if\;A_{n}<\frac{A_{1}+A_{2}+\ldots +A_{n}}{n}\\ F_{cc}=\left\{\left[\left|\left(\frac{A_{1}+A_{2}+\ldots +A_{n}}{n}\right)-A_{n}\right|÷A_{n}\right]\cdot F_{cn}\right\}-F_{cn};\;if\;A_{n}>\frac{A_{1}+A_{2}+\ldots +A_{n}}{n} \end{array}$$
where:

*A*_1+2+…+*n*_—*n*-th cross section of the stalk cut in the course of the *n*-th cutting test, with the specified values of *α*, *g* and *V*;

*F_cn_*—maximum value of the cutting force obtained from the *n*-th cutting test, with the specified values of *α*, *g* and *V*;

*F_cc_*—corrected maximum cutting force value during the cutting of the stalk, with the specified values of *α*, *g* and *V*;

*n—n*-th experimental attempt for the given values of *α*, *g* and *V*.

The experimental response values after applied correction were subjected to ANOVA analysis, and the results are presented below.

The results of the experimental study undergoing the correction discussed above are presented together with the input parameters (input values of the cutting process *α*—blade angle, *g*—distance between the blade and the counter-blade, *V*—linear velocity of the blade) in Table 2. This date obtained in the course of the experiment were also subject to ANOVA variance analysis. The corrected experimental response values were also subject to ANOVA analysis, as presented in Section 3. The values of total cutting energy, *E_tce_*, were determined by calculating the region under the characteristic curve of the cutting force, *F_c_*, measured in the course of the experiment, employing the trapezoidal rule to approximate the value of the integral (2) [30], which subsequently underwent a correction, similarly to force *F_c_*, according to the Formula (1). The specific energy value, *E_sce_*, was obtained by dividing the value, *E_tce_*, by the previously determined stalk cross-section area, *A* (see Figure 6):(2)$$P=dx\cdot \left(f_{1}+f_{2}+\ldots +f_{n-1}+\frac{f_{0}+f_{n}}{2}\right)$$
where:(3)$$dx=\frac{x_{k}-x_{p}}{n}$$

*P*—region under the characteristic curve of variance of the force *F_c_*;

*dx*—distance between the adjacent points of a given curve;

*x_k_—x_p_*—integration interval;

*n*—number of sections in which the curve was divided;

*f_i_* = *f(x_i_)*—value of the function at the point *x_i_*, for *i* = 0, 1, 2,…, *n*.

**Table 2 materials-16-03039-t002:** List of corrected average (from ten tests) maximum values of the initial parameters of the experiment expressed as the corrected cutting force, *F_cc_*, shear stress, *τ*, total cutting energy, *E_sce_*, and total cutting energy, *E_tce_*, for respective experimental input values.

Blade Angle *α* (°)	Distance *g* (mm)	Blade Velocity *V* (mm/s)	Cutting Force Value *F_cc_* (N)	Ultimate Shear Stress *τ*(MPa)	Total Cutting Energy *E_tce_* (J)	Specific Cutting Energy *E_sce_* (×10^−3^ J/mm^2^)
30	0.1	1	528.46	1.21	3.45	7.926
30	0.2	1	633.98	1.53	4.59	11.1
30	0.3	1	597.26	1.22	4.47	9.13
30	0.1	4	482.51	1.19	3.94	9.73
30	0.2	4	589.27	1.5	3.78	9.9
30	0.3	4	594.96	1.56	3.77	11.58
30	0.1	8	434.09	1.24	3.42	11.71
30	0.2	8	504.39	1.44	2.66	9.19
30	0.3	8	639.37	2	3.05	14.12
40	0.1	1	659.46	1.56	3.8	8.87
40	0.2	1	477.51	1.08	4.22	9.53
40	0.3	1	714.29	1.65	3.79	8.79
40	0.1	4	694.8	1.88	3.66	10.67
40	0.2	4	586.63	1.33	4.16	9.39
40	0.3	4	662.68	1.73	3.99	10.77
40	0.1	8	547.78	1.74	2.45	11.95
40	0.2	8	617.20	1.37	4.06	9
40	0.3	8	440.66	1.34	2.63	11.29
50	0.1	1	851.4	2.06	5.1	12.38
50	0.2	1	697.3	1.54	3.88	8.58
50	0.3	1	653.31	1.52	4.25	9.88
50	0.1	4	752.55	2.02	3.44	10.21
50	0.2	4	628.67	1.46	3.74	8.66
50	0.3	4	516.89	1.24	4.24	10.12
50	0.1	8	451.02	1.39	2.11	9.53
50	0.2	8	854.04	2.27	4.71	13.24
50	0.3	8	549.86	1.51	4.14	12.86
60	0.1	1	904.76	2.03	4.49	10.04
60	0.2	1	862.82	1.92	4.66	10.42
60	0.3	1	873.14	2.32	5.68	16.86
60	0.1	4	875.14	2.11	4.91	11.87
60	0.2	4	735.25	1.94	4.29	11.98
60	0.3	4	837.94	2.44	3.58	13.35
60	0.1	8	726.35	1.95	4.48	12.9
60	0.2	8	561.21	1.65	3.38	12.78
60	0.3	8	814.95	2.53	2.98	13.09
70	0.1	1	1353.3	3.38	5.71	14.29
70	0.2	1	965.33	3.04	3.92	19.17
70	0.3	1	1060.63	2.57	4.73	11.44
70	0.1	4	1147	3.21	3.71	12.18
70	0.2	4	940.29	3.12	2.66	16.56
70	0.3	4	688.04	1.78	3.83	10.38
70	0.1	8	855.4	2.72	2.52	12.57
70	0.2	8	838.71	2.99	2.29	17.34
70	0.3	8	1061.4	3.41	3.44	17.78
80	0.1	1	1182.33	2.43	5.59	11.51
80	0.2	1	1362.19	3.31	5.49	13.37
80	0.3	1	1151.31	2.89	5.92	14.94
80	0.1	4	1134.41	2.49	5.02	11.04
80	0.2	4	1231.16	3.15	4.84	12.36
80	0.3	4	1097.95	3.06	3.94	13.02
80	0.1	8	1629.539	5.31	4.37	24.27
80	0.2	8	1039.44	2.97	3.79	13.18
80	0.3	8	929.09	3.02	2.94	15.91

### 3.1. Multivariate Analysis of Cutting Force, F_cc_

This section describes the ANOVA analysis carried out for the maximum cutting force, *F_cc_*. Its value was determined on the basis of the graph line of the variance of the cutting force measured in the course of the experimental study of the corn-stalk cutting process, depending on the blade angle, *α* (°), and the distance between the blade and the counter-blade, *g* (mm), as well as the linear velocity of the blade *V* (mm/s). Table 3 presents the results of the ANOVA analysis. The analysis employed a reduced quadratic model, for which R^2^ = 0.8116. According to Table 3, the statistical F-value of the model is 21.07, which means the model is significant, whereas the *p*-value for the model components, i.e., blade angle, *α*, blade velocity, *V*, the product of the angle value, *α*, and distance, *g,* and the squared value of the angle, *α*^2^, is lower than 0.05, which means that these components are significant. The parameter values are predicted R^2^ = 0.6860 and adjusted R^2^ = 0.7731, and the difference between these values is lower than 0.2, which, in combination with the Adeq Precision value of 17.1057, signifies that the model is usable within the design framework of the experiment. The model is expressed with the dependence (4). The model graph line is provided in Figure 7.
(4)$$F_{cc}=633.614+\left(-6.24546\cdot \alpha \right)+344.845\cdot g+\left(-36.2416\cdot V\right)+\left(-32.3887\cdot \alpha \cdot g\right)+\;(-0.0937412\cdot \alpha \cdot V)+30.2669\cdot g\cdot V+0.236325\cdot \alpha^{2}+2342.17\cdot g^{2}+2.13318\cdot V^{2}$$

*F_cc_*—cutting force (N), *α*—knife angle (°), *g*—distance between the blade and the counter blade (mm), *V*—knife blade velocity (mm/s).

**Table 3 materials-16-03039-t003:** ANOVA results. Dependent variable—cutting force—*F_cc_* (N).

Source	Sum of Squares	df ^a^	Mean Square	F-Value	*p*-Value	
Model	3.083 × 10^6^	9	3.426 × 10^5^	21.07	<0.0001	significant
*α*	2.592 × 10^6^	1	2.592 × 10^6^	159.41	<0.0001	
*g*	47,397.54	1	47,397.54	2.91	0.0948	
*V*	1.150 × 10^5^	1	1.150 × 10^5^	7.07	0.0109	
*α g*	1.101× 10^5^	1	1.101 × 10^5^	6.77	0.0126	
*α V*	1137.97	1	1137.97	0.0700	0.7926	
*g V*	2711.62	1	2711.62	0.1668	0.6850	
*α^2^*	1.877 × 10^5^	1	1.877 × 10^5^	11.54	0.0015	
*g^2^*	6582.91	1	6582.91	0.4048	0.5279	
*V^2^*	7810.06	1	7810.06	0.4803	0.4919	

**^a^** degrees of freedom.

**Figure 7 materials-16-03039-f007:**
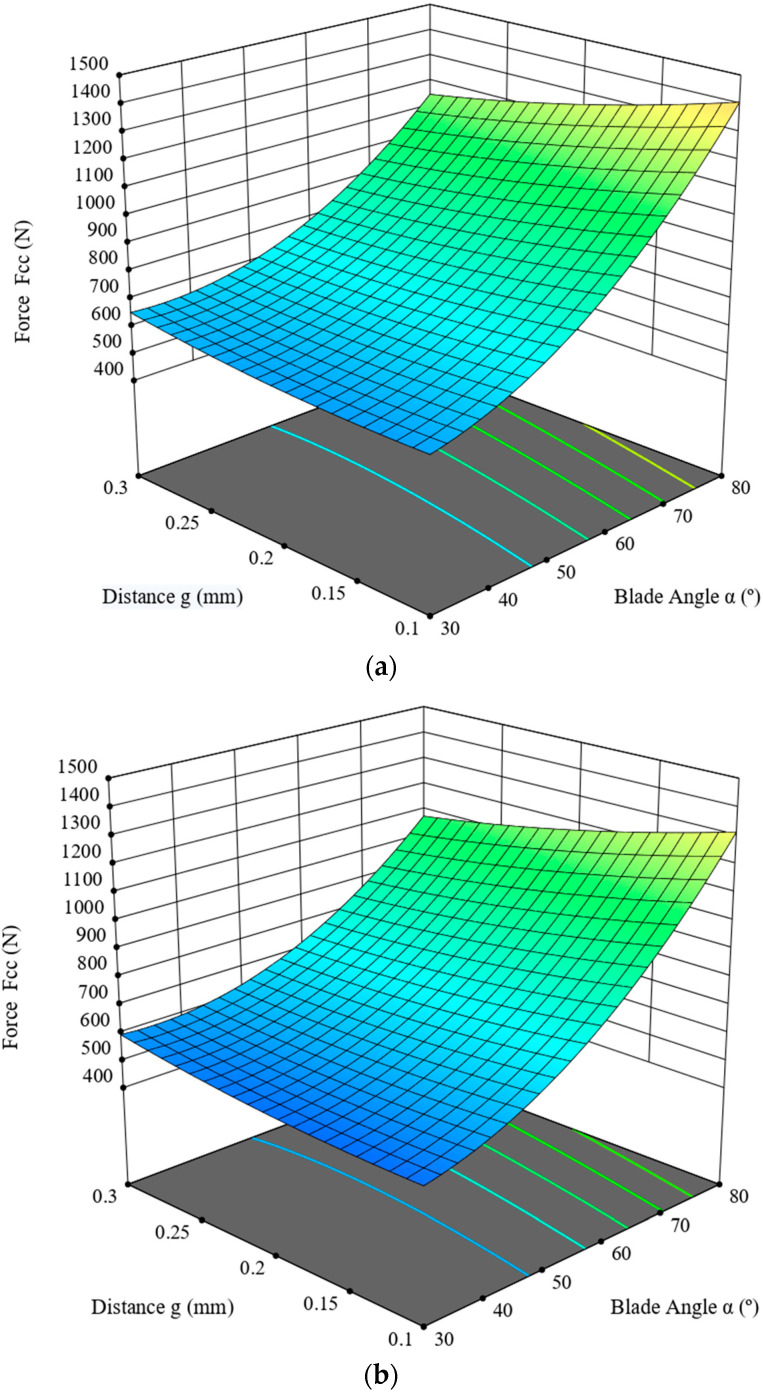
Cutting force value, *F_cc_,* in the function of the blade angle, *α*, of the distance, *g,* for velocity value (**a**) *V* = 1 mm/s, (**b**) *V* = 4 mm/s and (**c**) *V* = 8 mm/s.

As follows from the data provided in Table 3, the highest value (F = 159.41, see Table 3) was measured for the experiment variable of the blade angle, *α*, which has the largest influence on the value of the cutting force. The second most important parameter influencing this value is the linear velocity of the blade, *V;* in this case, the value is F = 7.07. The cutting-force value decreases significantly, together with the decrease in the blade angle, *α*, and the increase in its linear velocity, *V.* For the three characteristics mentioned above, it is evident that the influence of the distance value, *g*, is minimal for the blade-angle value range of 30–60°. For higher values, the influence is more pronounced.

### 3.2. Multivariate Analysis of Ultimate Shear Stress, τ

This section is devoted to the ANOVA analysis for the maximum value of the ultimate shear stress, *τ* (MPa). It was determined on the basis of the graph line of the variance of the cutting force measured in the course of the experimental study of the corn-stalk cutting process, depending on the blade angle, *α* (°), the distance between the blade and the counter-blade, *g* (mm), as well as the linear velocity of the blade, *V* (mm/s). Table 4 presents the results of the ANOVA analysis. The analysis employed a linear model, for which R^2^ = 0.6580. According to Table 4, the statistical F-value of the model is 32.06, which means the model is significant, whereas the *p*-value for the blade angle, *α*, is lower than 0.05, denoting that this value constitutes a significant component thereof. The parameter values are predicted R^2^ = 0.5880 and adjusted R^2^ = 0.6374, and the difference between these values is lower than 0.2, which, in combination with the Adeq Precision value of 16.6224, signifies that the model is usable within the design framework of the experiment. The model is expressed with the dependence (5). The model graph line is provided in Figure 8.
(5)τ=0.0232326+0.0381565·α+(−0.583048·g)+0.0299192·V

*τ*—ultimate shear stress (MPa), *α*—knife angle (°), *g*—distance between the blade and the counter blade (mm), *V*—knife blade velocity (mm/s)

As follows from the data provided in Table 4, the highest value (F = 94.05, see Table 4) was measured for the experimental variable of the blade angle, *α*, which has the largest influence on the value of the cutting force. The second most important parameter influencing this value is the linear velocity of the blade, *V*; in this case, the value is F = 1.63. The next most important parameter is the distance *g* between the blade and the counter-blade. The cutting-force value decreases significantly, together with the decrease in the blade angle, *α*, and the decrease in its linear velocity, *V.* For the three characteristics mentioned above, it is evident that the influence of the distance value, *g*, is minimal.

### 3.3. Multivariate Analysis of Total Cutting Energy, E_tce_

This section is devoted to the ANOVA analysis for the maximum value of the total cutting energy, *E_tce_* (J). It was determined on the basis of the graph line of the variance of the cutting force measured in the course of the experimental study of the corn-stalk cutting process, depending on the blade angle, *α* (°), the distance between the blade and the counter-blade, *g* (mm), as well as the linear velocity of the blade, *V* (mm/s). Table 5 presents the results of the ANOVA analysis. The analysis employed a reduced 2FI model, a two-factor interaction linear model, for which R^2^ = 0.4204. According to Table 5, the statistical F-value of the model is 12.09, which means the model is significant, whereas the *p*-value for the blade angle, *α*, is lower than 0.05, denoting that this value constitutes a significant component thereof. The parameter values are predicted R^2^ = 0.3054 and adjusted R^2^ = 0.3856, and the difference between these values is lower than 0.2, which, in combination with the Adeq Precision value of 11.5761, signifies that the model is usable within the design framework of the experiment. The model is expressed with the dependence (6). The model graph line is provided in Figure 9.
(6)$$E_{tce}=1.70871+0.0497732\cdot \alpha +6.37737\cdot g+\left(-0.107243\cdot \alpha \cdot g\right)$$

*E_tce_*—total cutting energy (J), *α*—knife angle (°), *g*—distance between the blade and the counter blade (mm).

As follows from the data provided in Table 5, the highest value (F = 32.90, see Table 5) was measured for the experiment variable of the blade angle, *α*, which has the largest influence on the value of the cutting force. The second most important parameter is the interaction between the angle value, *α*, and distance, *g*; in this case, the value is F = 3.14.

### 3.4. Multivariate Analysis of Specific Cutting Energy, E_sce_

This section is devoted to the ANOVA analysis for the maximum value of the specific cutting energy, *E_sce_* (J/mm^2^) of the shear stress, *τ*. It was determined on the basis of the graph line of the variance of the cutting force measured in the course of the experimental study of the corn-stalk cutting process, depending on the blade angle, *α* (°), the distance between the blade and the counter-blade, *g* (mm,) as well as the linear velocity of the blade, *V* (mm/s). Table 5 presents the results of the ANOVA analysis. The analysis employed a linear model, for which R^2^ = 0.4091. According to Table 6, the statistical F-value of the model is 11.54, which means the model is significant, whereas the *p*-value for the blade angle, *α*, is lower than 0.05, denoting that this value constitutes a significant component thereof. The parameter values are predicted R^2^ = 0.2965 and adjusted R^2^ = 0.3737, and the difference between these values is lower than 0.2, which, in combination with the Adeq Precision value of 11.7537, signifies that the model is usable within the design framework of the experiment. The model is expressed with the dependence (7). The model graph line is provided in Figure 10.
(7)$$E_{sce}=4.66959+0.101064\cdot \alpha +3.23599\cdot g+0.288363\cdot V$$

*E_sce_*—total cutting energy (J/mm^2^), *α*—knife angle (°), *g*—distance between the blade and the counter blade (mm), *V*—knife blade velocity (mm/s)

As follows from the data provided in Table 6, the highest value (F = 27.63, see Table 6) was measured for the experiment variable of the blade angle, *α*, which has the greatest influence on the value of the cutting force. The second most important parameter is the linear velocity of the blade, *V*; in this case, the value is F = 6.34. The cutting-force value decreases significantly, together with the decrease in the blade angle, *α*, and the increase in its linear velocity, *V.* A significant downward trend is observed for the value of *E_sce_* together with the decrease in the distance, *g*.

## 4. Discussion

Table 7 provides the hierarchy of influence of the parameters based on the F-value, the components and interactions (*α*—blade angle (°), *g*—distance between the blade and the counter-blade (mm), *V*—linear velocity of the blade (mm/s)) for the model representing the experiment in relation to the analyzed response, i.e., the corn-stalk cutting force, *F_cc_*.

As provided in Table 7, the responses were obtained in the course of the cycle of studies (Section 2 and Section 3) of cutting corn stalks harvested from the fields located in the Wielkopolska Province (west-central Poland); the value of the corn-stalk cutting force, *F_cc_* is primarily influenced by the blade angle, *α*, (Table 7) and this variable is the significant component of the model expressed in the form of Equation (4) in Section 3.1, as signified by the *p*-value of < 0.0001. Similarly, the studies of Wang et al. [22] also demonstrated that the blade angle and its interaction with the distance between the blade and the counter blade (see Table 7) has the greatest influence on the process of energy consumption. Moreover, as provided in Figure 7, together with the increase in the linear velocity of the blade, *V,* we observe a decrease in the force value, *F_cc_*, for the entire range of values of the distance, *g,* and the blade angle, *α.* The lowest force value was recorded for the blade angle *α =* 30°, the distance *g =* 0.1 mm and the blade linear velocity *V* = 8 mm/s. The value of shear stress *τ* (see Table 7) is influenced the most by the blade angle, *α* (*p* < 0.0001). Moreover, with the increase in the linear velocity of the blade, *V*, a minimal increase in the shear stress *τ* value is observed for the entire range of the value of distance, *g*, and the blade angle, *α.* The lowest stress value, *τ*, is measured for the blade angle *α =* 30°, the distance *g =* 0.1 mm and the linear velocity of the blade *V* = 1 mm/s (see Figure 8). The total cutting energy value, *E_tce_*, is not influenced by the linear velocity, *V,* of the blade, but the blade angle, *α,* which at the same time constitutes a significant component of the model expressed as (6). The lowest value of the *E_tce_* was achieved for the blade angle *α =* 30°, and distance *g =* 0.1 mm (Figure 9). The specific cutting energy, *E_sce_*, is primarily dependent on the blade-angle value, *α* (see Table 7), and this angle is also considered a significant component of the model (*p* < 0.0001) expressed as (7). Second in the hierarchy of significance is the linear velocity value, *V* (see Table 7); the increase in this value causes an increase in *E_sce_* for the entire range of values of the distance, *g*, and the blade angle, *α* (see Figure 10). The lowest *E_sce_* value was measured for the blade angle *α =* 30°, distance *g =* 0.1 mm and the linear blade velocity *V* = 1 mm/s (Figure 10).

### 4.1. Influence of the Geometrical Features of the Knife on Its Life

The distribution of forces on the blade was analyzed with view to determining the correlations between individual forces as well as their variance influenced by the value of the angle, *α* (see Figure 11). This analysis is to determine the correlation between blade strength and its geometrical features, specifically the angle, *α*. The authors refer to numerous examples in the subject literature in which the correlation between the wear of the tool and the cutting forces are presented [31,32,33,34,35,36]. As tool life is dependent on frictional wear, which can be estimated by measuring the radius of the milling edge, *r* (in the case of the knife cutting the stalk, this refers to its cutting edge), it was assumed that the value of the frictional force will result in faster wear and reduce the life of the tool. This assumption is based on the work [37], whose authors have determined that increased tool wear is caused by an increase in the contact area and the increase in the coefficient of friction. This observation was referred to the blade used in the present examination of the cutting process of the corn stalk. The ratio of forces, *F_cc_*/*T_x_*, was determined based on the performed load state analysis in the second stage of cutting the corn stalk (see Figure 4 and Figure 5), during which the corn stalk compressed in the first stage of the process is separated. The value of this ratio will serve as an indicator of the life of the blade cutting the stalk. Figure 11 presents the distribution of forces on the knife in order to determine the frictional force component, *T_x_*.

As a reminder, the cutting force measured in the course of the study is denoted as *F_c_* (see Figure 4), whereas its corrected value according to the dependence (1) (Section 2.2) is denoted as *F_cc_*. Based on the analyzed distribution of forces, the horizontal component, *T_x_*, of the frictional force, *T*, (see Figure 11) was established according to the Formula (8):(8)$$\frac{T_{x}}{T}=\sin\alpha$$
where:

*T_x_*—horizontal frictional force component *T* (N);

*T*—frictional force (N);

*α*—blade angle (°).

Whereas the formula to calculate the frictional force, *T*, can be expressed as follows:(9)$$T=F_{n}\cdot \mu$$
where:

*T*—frictional force (N);

*F_n_*—normal force of the frictional force *T* (N);

*μ*—coefficient of friction according to Richter (1954) *μ* = 0.82 [38].

The dependence for calculating the normal force of the frictional force is expressed in the following form (10) (see Figure 11):(10)$$\frac{F_{n}}{F_{cc}}=\cos\left(90{}^{\circ}-\alpha \right)$$
therefore:(11)$$F_{n}=F_{cc}\cdot \cos\left(90{}^{\circ}-\alpha \right)$$

The dependence for calculating the frictional force, *T*, can be expressed as follows:(12)$$T=F_{cc}\cdot \cos\cdot \left(90{}^{\circ}-\alpha \right)\cdot \mu$$

Hence, the horizontal component, *T_x_*, of the frictional force, *T*, can eventually be expressed as follows (13):(13)$$T_{x}=F_{cc}\cdot \cos\cdot \left(90{}^{\circ}-\alpha \right)\cdot \mu \cdot \sin\alpha$$

This was used to determine the value of the ratio of the cutting force, *F_cc_*, and the horizontal frictional force component, *T_x_*, *F_cc_/T_x_*. In Figure 12, the distribution of values of this ratio was presented in the function of the blade angle, *α*.

It needs to be emphasized that the values of the ratio *F_cc_/T_x_* are identical for the different values of the distance between the blade and the counter blade and the linear velocity value, *V*, for a single value of the blade angle, *α*. According to the above characteristic, and at the same time based on the assumption presented in the work [37], it is possible to postulate that the lower the value of the *F_cc_/T_x_* ratio, the greater the wear life of the knife used for cutting the corn stalk.

### 4.2. Optimisation of the Selection of Cutting Process Parameters

Based on the considerations provided in Section 4.1, which resulted in formulating the correlation between the cutting force, *F_cc_*, and the horizontal component of the frictional force, *T_x_*, the characteristic as provided in Figure 12 was developed. This facilitated a discussion on the possible optimization, which considered two functions. The first is the characterization of variance in the function of variance of the blade angle, *α*, and the distance between the blade and the counter-blade, *g*, for the linear-velocity value *V* = 8 mm/s, since at this velocity value the cutting force in the entire range of the blade angle and distance values was lower in comparison to the other two velocity values, *V* = 1 mm/s and *V* = 4 mm/s. The second consideration is the characteristic curve as provided in Figure 12. This was used to establish the optimization criteria, as provided in Table 8.

The results of the optimization are provided in Table 9, in function of the variable weight value for the force value, *F_cc_*, and the criterion of the ratio *F_cc_/T_x_*. The weight was adjusted by increasing it for the value of the force *F_cc_* and simultaneously reducing the weight by the same amount for the criterion of the *F_cc_*/*T_x_* ratio. The increase in the weight for the force value, *F_cc_*, represents an increased tendency to prioritize the minimization of the cutting force.

The above table presents the values determined on the basis of the optimization process with the criteria in Table 8. These results allow us to conclude that the optimal blade angle, *α,* is in the range of 40–60°. The exact value from the range would depend on the importance of minimizing the cutting force. In this case, for a weight value of 0.9, the force value, *F_cc_*, achieves the minimum value, thus minimizing the required energy to separate the corn stalk. However, this is at the expense of the blade life. However, if the blade is made of a high-quality material with good strength and wear resistance, the blade angle of *α* = 40–42° may be considered. However, if it is not economically feasible to manufacture the blade from steel with good mechanical characterization, a lower-quality material is feasible for the blade angle of *α* = 60°, thus causing an increase in the blade life but also increasing the force value required to separate the corn stalk. For the entire range of weight values, the value, *g*, varies in the range from 0.3 for the value of 0.5/0.5 to approx. 0.2 for the weight values 0.9/0.1 (see Table 9). The linear velocity of the blade knife for the entire range of weight variance is equal to *V* = 8 mm/s. Correspondingly, the blade-life value is highest for the weight value of 0.5/0.5, and is equal to *F_cc_*/*T_x_* = 1.47, and the lowest for the weight value of 0.9/0.1, and is equal to *F_cc_*/*T_x_* = 2.96.

## 5. Conclusions

Based on the results of the experimental study and the following analyses and optimization, it can be said that the correction of the cutting force, *F_c_*, which was measured directly in the course of the experimental study allowed for the elimination of the negative influence of the variable cross-sectional area of the corn stalk on the analysis of the influence on the cutting process and the interaction of the specific input parameters of the cutting process. The ANOVA variance analysis of the obtained experimental results demonstrates that the lowest force value, *F_cc_*, required to separate the corn stalk is obtainable for *α* = 30°, *g* = 0.1 mm and *V* = 8 mm/s. Considering the shear stresses, *τ*, these correspond to parameter values of *α* = 30°, *g* = 0.1 mm and *V* = 1 mm/s. For the total cutting energy, *E_tce_*, these are the values of *α* = 30°, *g* = 0.1 mm, regardless of the linear velocity of the blade, *V*. For the specific cutting energy, *E_sce_*, the input settings for the corn-stalk cutting process are respectively *α* = 30°, *g* = 0.1 mm and *V* = 1 mm/s. Considering the analysis of distribution of the forces on the knife blade (Section 4.1), which leads to the determination of the force ratio, *F_cc_*/*T_x_*, on the basis of the assumed criterion from the publication [37], the process of optimization allows us to identify the optimal blade angle, *α*, from the range of values of 40–60°. The actual angle value depends on the weights assumed for the optimization process. Considering the extreme weight values assumed for the optimization criteria (see Table 8), the optimal blade-angle value is 60°, with weights of 0.5 and 0.5 respectively for the *F_cc_* and the *F_cc_*/*T_x_* ratio, whereas if the most important factor is the minimization of the cutting force in order to reduce the process of energy consumption of the corn-stalk cutting process, assuming a weight of 0.9 for the *F_cc_* and 0.1 for the *F_cc_*/*T_x_* ratio, the optimal blade-angle value is 40°.

## Figures and Tables

**Figure 1 materials-16-03039-f001:**
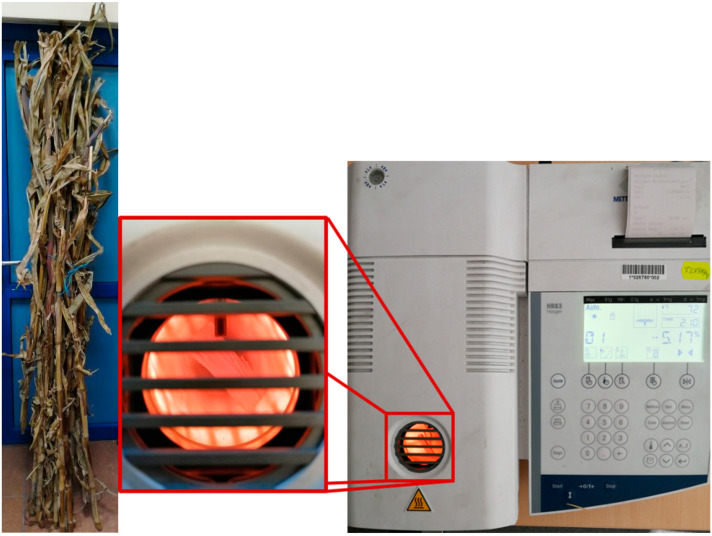
General view of the seasoned corn stalks (**left**), determining the moisture content of the corn stalks using a scale-dryer (**right**).

**Figure 2 materials-16-03039-f002:**
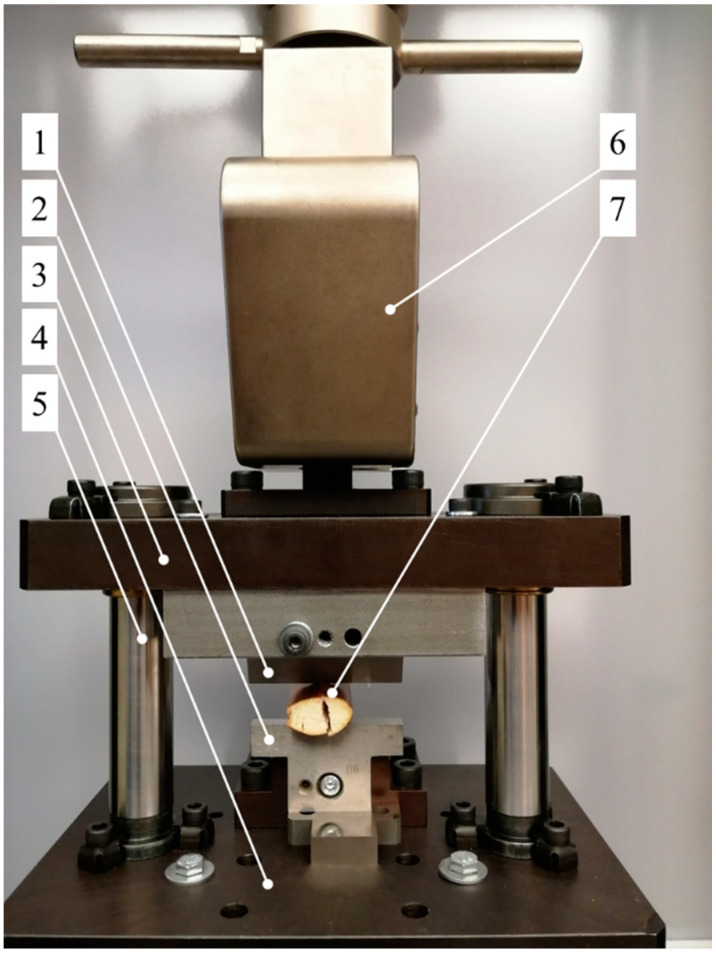
Schematic view describing the construction of the testing station employed in the study of the corn-stalk cutting process, where: 1—knife blade, 2—counter-blade, 3—upper plate, 4—lower plate, 5—roller guide, 6—jaws of the MTS testing machine, 7—corn stalk subject to cutting.

**Figure 3 materials-16-03039-f003:**
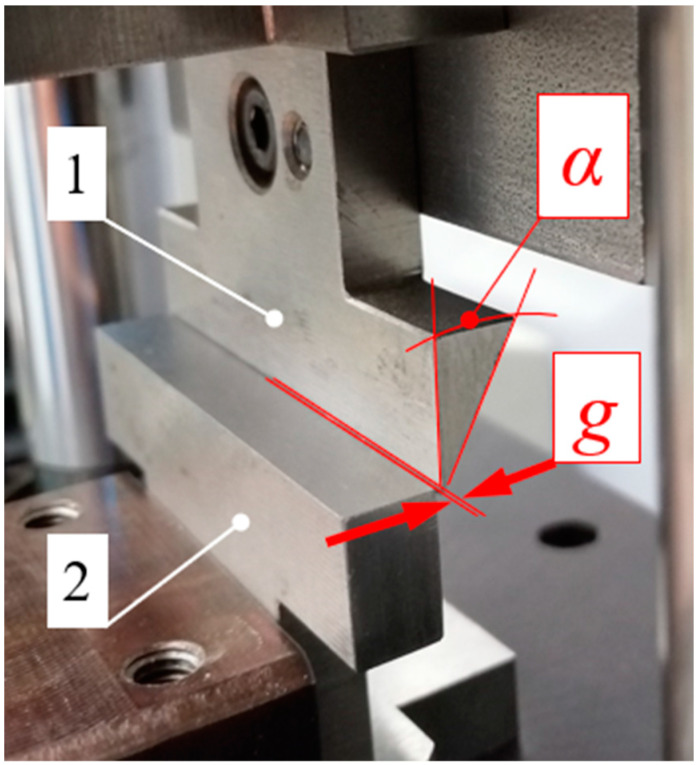
General view of the blade angle, *α*, and the distance g between the blade, 1, and the counter-blade, 2, of the testing station employed for cutting the corn stalks, with a general view of the blades employed in the study with the range of the blade-angle value α = 30–80°.

**Figure 4 materials-16-03039-f004:**
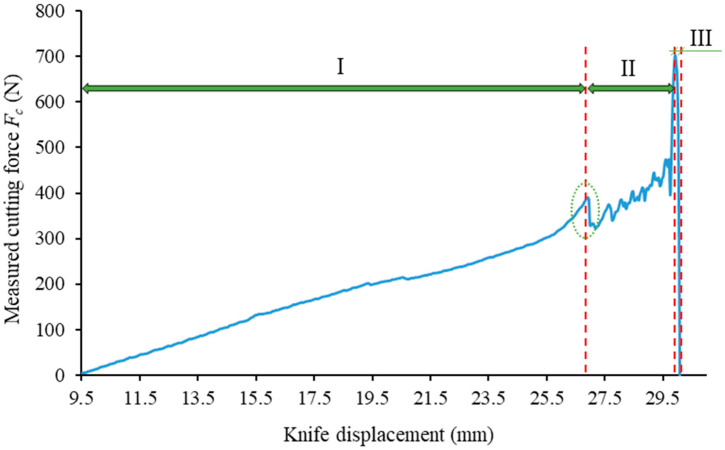
Example characteristic curve showing the variance in the measured cutting force, *F_c_*, of the corn stalk for the parameter values: *α* = 40°, *g* = 0.1 mm and *V* = 1 mm/s.

**Figure 5 materials-16-03039-f005:**
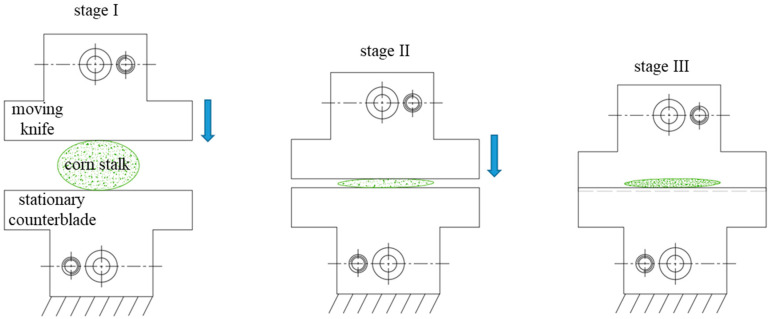
Stages of the corn-stalk cutting process.

**Figure 6 materials-16-03039-f006:**
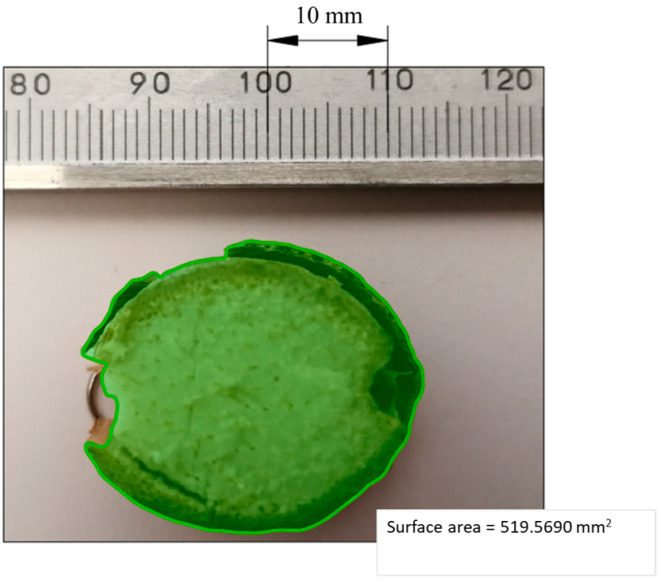
Method of determining the cross-sectional area of the corn stalk in the AutoCad software.

**Figure 8 materials-16-03039-f008:**
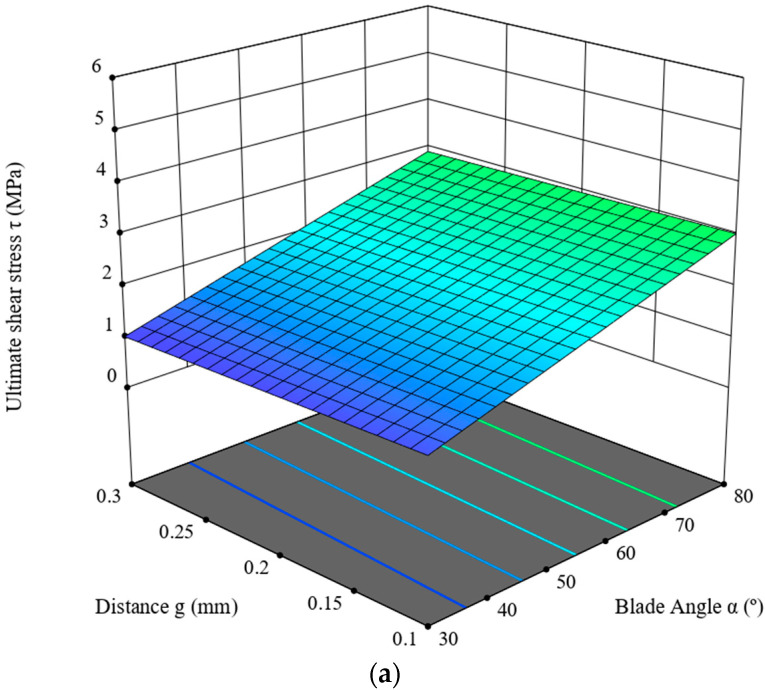
Ultimate shear stress, *τ,* in the function of the blade angle, *α*, of the distance, *g,* for velocity value (**a**) *V* = 1 mm/s, (**b**) *V* = 4 mm/s and (**c**) *V* = 8 mm/s.

**Figure 9 materials-16-03039-f009:**
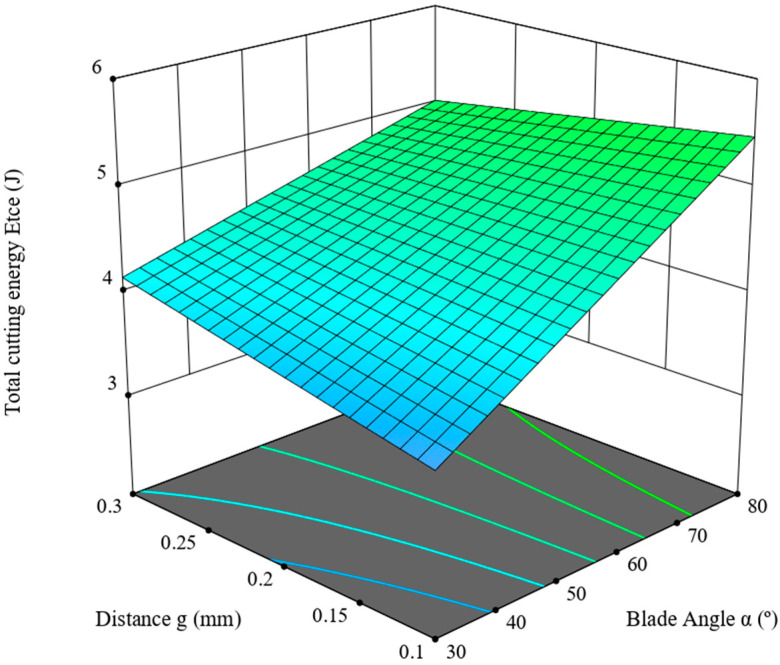
Total cutting energy, *E_tce_,* in the function of the blade angle, *α*, of the distance, *g*.

**Figure 10 materials-16-03039-f010:**
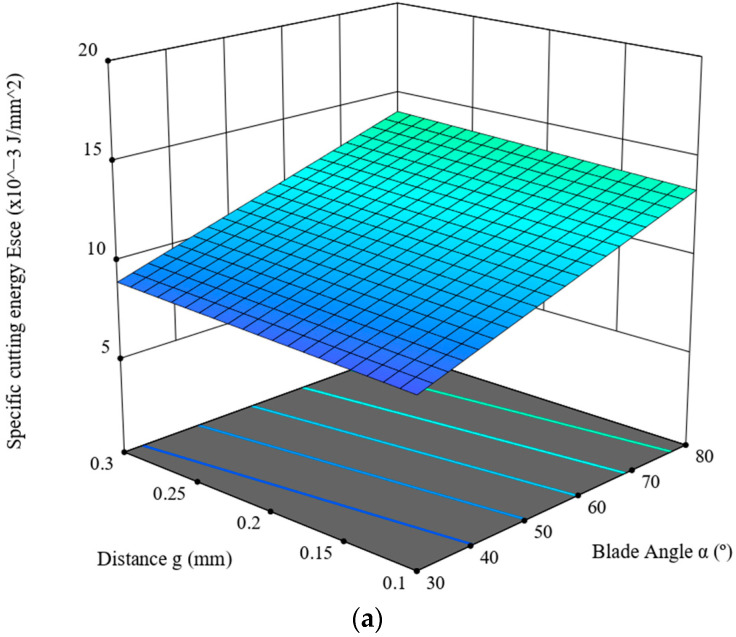
Specific cutting energy, *E_sce_,* in the function of the blade angle, *α*, of the distance, *g,* for velocity value (**a**) *V* = 1 mm/s, (**b**) *V* = 4 mm/s and (**c**) *V* = 8 mm/s.

**Figure 11 materials-16-03039-f011:**
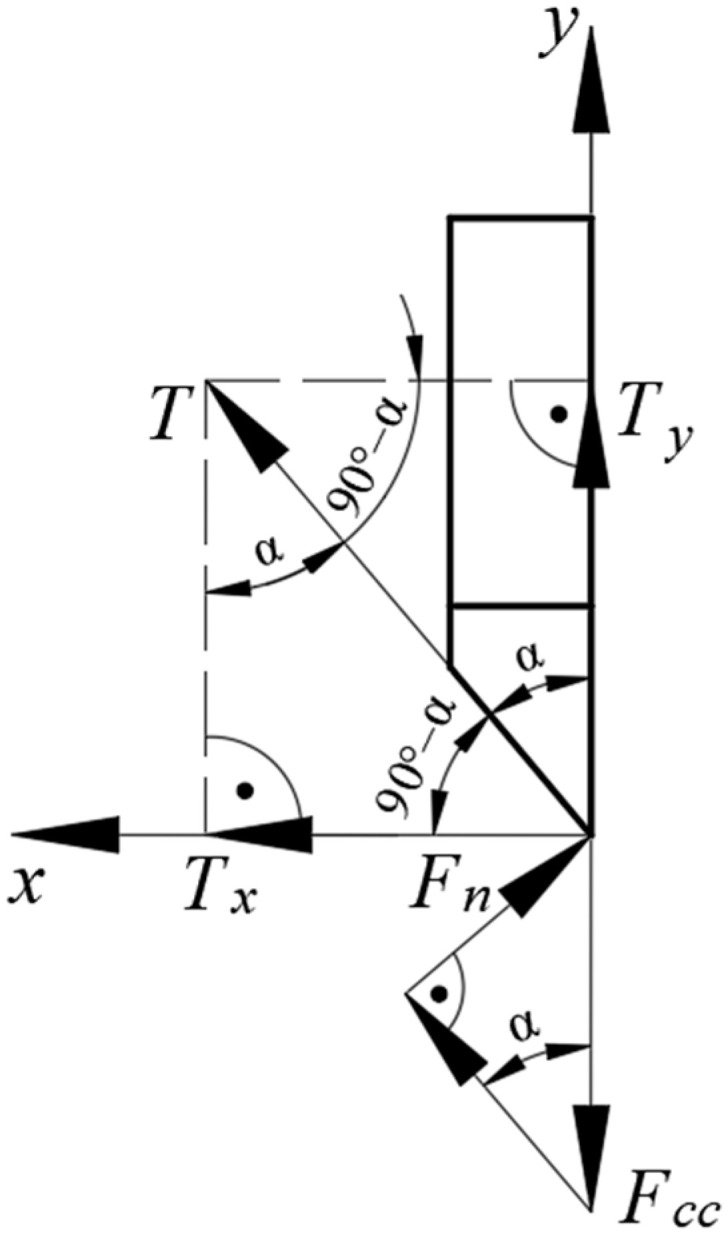
The analyzed distribution of forces on the knife blade, where: *F_cc_*—corrected value of the measured cutting force (N), *T*—frictional force (N), *T_y_*—vertical frictional force component (N), *T_x_*—horizontal frictional force component (N), *F_n_*—normal force of the frictional force (N), *α*—blade angle (°).

**Figure 12 materials-16-03039-f012:**
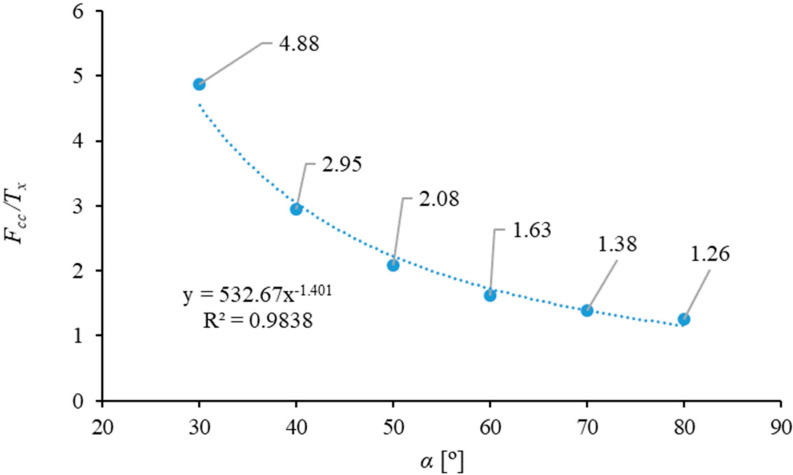
Characteristic curve representing the variance of the ratio of forces *F_cc_/T_x_*.

**Table 1 materials-16-03039-t001:** Example values of the corn stalk cross-sectional area in elliptical shape for the blade angle *α* = 30°, distance *g* = 0.1, 0.2, 0.3 mm and blade velocity *V* = 1 mm/s, 4 mm/s and 8 mm/s.

Blade Angle *α* (°)	Distance *g* (mm)	Blade Velocity *V* (mm/s)	Corn Stalk Cross-Sectional Area *A* (mm^2^)
30	0.1	1	516.90
30	0.2	1	607.75
30	0.3	1	689.45
30	0.1	4	442.39
30	0.2	4	578.69
30	0.3	4	594.76
30	0.1	8	322.86
30	0.2	8	347.96
30	0.3	8	410.54

**Table 4 materials-16-03039-t004:** ANOVA results. Dependent variable—ultimate shear stress—*τ* (MPa).

Source	Sum of Squares	df ^a^	Mean Square	F-Value	*p*-Value	
Model	23.45	3	7.82	32.06	<0.0001	significant
*α*	22.93	1	22.93	94.05	<0.0001	
*g*	0.1224	1	0.1224	0.5019	0.4819	
*V*	0.3975	1	0.3975	1.63	0.2076	

**^a^** degrees of freedom.

**Table 5 materials-16-03039-t005:** ANOVA results. Dependent variable—total cutting energy—*E_tce_* (J).

Source	Sum of Squares	df ^a^	Mean Square	F-Value	*p*-Value	
Model	13.93	3	4.64	12.09	<0.0001	significant
*α*	12.64	1	12.64	32.90	<0.0001	
*g*	0.0826	1	0.0826	0.2151	0.6448	
*α g*	1.21	1	1.21	3.14	0.0823	

**^a^** degrees of freedom.

**Table 6 materials-16-03039-t006:** ANOVA results. Dependent variable—specific cutting energy—*E_sce_* (J/mm^2^).

Source	Sum of Squares	df ^a^	Mean Square	F-Value	*p*-Value	
Model	201.56	3	67.19	11.54	<0.0001	significant
*α*	160.87	1	160.87	27.63	<0.0001	
*g*	3.77	1	3.77	0.6475	0.4248	
*V*	36.92	1	36.92	6.34	0.0150	

**^a^** degrees of freedom.

**Table 7 materials-16-03039-t007:** Hierarchy of components of individual models based on the F-value criterion. The F-value is provided in brackets.

Response	1	2	3	4	5
*F_cc_*	*α* (159.41)	*α^2^* (11.54)	*V* (7.07)	*α × g* (6.77)	*g* (2.91)
*τ*	*α* (94.05)	*V* (1.63)	*g* (0.5019)		
*E_tce_*	*α* (32.90)	*α × g* (3.14)	*g* (0.2151)		
*E_sce_*	*α* (27.63)	*V* (6.34)	*g* (0.6475)		

**Table 8 materials-16-03039-t008:** Assumed criteria of optimization.

Input/Output Variable	Goal	Lower Limit	Upper Limit
*α* (°)	is in range	30	90
*g* (mm)	is in range	0.1	0.3
*V* (mm/s)	is in range	1	8
*F_cc_* (N)	minimize	434.1	1629.5
*F_cc_*/*T_x_* (-)	minimize	1.26	4.88

**Table 9 materials-16-03039-t009:** Process parameters meeting the criteria for digital optimization.

Weight *F_cc_*/(*F_cc_*/*T_x_*)	Blade Angle *α* (°)	*g* (mm)	*V* (mm/s)	*F_cc_* (N)	*F_cc_*/*T_x_*
0.5/0.5	60.2	0.29	7.74	717.4	1.47
0.6/0.4	57.1	0.27	7.82	681.4	1.63
0.7/0.3	53.4	0.24	7.94	640.7	1.88
0.8/0.2	48.7	0.21	7.99	595	2.27
0.9/0.1	42.1	0.17	7.99	540.4	2.96

## Data Availability

The data presented in this study are available on request from the corresponding author.

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
