# Peer review of "Experimental Study on the Mechanical Behavior of Dry Corn Stalk Cutting"

_materials, 2023, doi:10.3390/ma16083039_

Round 1

Reviewer 1 Report

minor revision 

Author Response

Dear Reviewer,

I would like to thank you for the time devoted to the review of this article. All the comments are very valuable. The authors made an effort to account for all the remarks with view of a positive review of the corrected version of the manuscript. The authors would furthermore like to express their hopes that the responses and corrections made to the article are satisfactory and will enable its publication in Materials.

Please see below for our responses (discussion) to your individual remarks.

Reviewer 2 Report

This paper used ANOVA tools to analyze the interactions between blade angle, blade speed and distance and response quantities, which obtained optimized values for reducing the energy consumption of the corn stalk cutting process. It has presented appreciable work. However, for enhancing its quality following suggestions must be taken into account.

.1. The background of the article is detailed and informative, but it is recommended to simplify the background introduction related to non-fibrous materials.

2. Some details of the equations in the article need to be corrected, including Eqs(1),(10) and (11).

3. A more detailed explanation of the relationship between the response volume and the shaping of biomass particles for the cutting process in the article would be helpful.

4. In line 257, how is the forming process of corn stover? Is there a guarantee that the material properties are uniform? Will there be pre-damage to the material from the processing?

5. How are the cutting parameters in Table 2 of the article selected? Are they reliable for actual production processing?

Author Response

(The authors gave the same response as above.)

Reviewer 3 Report

This research aims to determine the highest possible energy concentration together with the highest possible resistance to mechanical damage of corn. The problem is clearly explained and the authors have provided a good literature review and discussion of existing work. the methods are fully explained and the research results are well discussed. However, there are some comments: -  Pages 40-43 and 49-51. Sentences are repeated. please, properly cite the kinds of literature all over the article. For example, Zastempowski and Bochat, year of publication (page 62). In this case, there is no need for [15] (page 72). It is hard to understand what is the alfa angle in figure 3. please show this angle clearly and how this factor was controlled in this figure or provide more figures from various sides. - Figures 4,5 and 6, as a result of the experiment should be written in the result part. There is no need to show table 1 as an example. If there are experiment results better to write in the results part.  -pages 560-602 should be written in the methods and materials part as long as the discussion part contains experiment results. I suggest combining the research and discussion parts.  

Author Response

(The authors gave the same response as above.)

Round 2

Reviewer 3 Report

Much better, It would be better to organize figure 3